# Sparse Linear Programming via Primal and Dual Augmented Coordinate Descent

**Ian E.H. Yen** [*]    **Kai Zhong** [*]    **Cho-Jui Hsieh** [†]    **Pradeep Ravikumar** [*]    **Inderjit S. Dhillon** [*]

[*] University of Texas at Austin        [†] University of California at Davis

[*] {ianyen,pradeepr,inderjit}@cs.utexas.edu   zhongkai@ices.utexas.edu

[†] chohsieh@ucdavis.edu

## Abstract

Over the past decades, Linear Programming (LP) has been widely used in different areas and considered as one of the mature technologies in numerical optimization. However, the complexity offered by state-of-the-art algorithms (i.e. interior-point method and primal, dual simplex methods) is still unsatisfactory for problems in machine learning with huge number of variables and constraints. In this paper, we investigate a general LP algorithm based on the combination of Augmented Lagrangian and Coordinate Descent (AL-CD), giving an iteration complexity of $O((\log(1/\epsilon))^2)$ with $O(nnz(A))$ cost per iteration, where $nnz(A)$ is the number of non-zeros in the $m \times n$ constraint matrix $A$, and in practice, one can further reduce cost per iteration to the order of non-zeros in columns (rows) corresponding to the active primal (dual) variables through an active-set strategy. The algorithm thus yields a tractable alternative to standard LP methods for large-scale problems of sparse solutions and $nnz(A) \ll mn$. We conduct experiments on large-scale LP instances from $\ell_1$-regularized multi-class SVM, Sparse Inverse Covariance Estimation, and Nonnegative Matrix Factorization, where the proposed approach finds solutions of $10^{-3}$ precision orders of magnitude faster than state-of-the-art implementations of interior-point and simplex methods. [1]

## 1   Introduction

Linear Programming (LP) has been studied since the early 19th century and has become one of the representative tools of numerical optimization with wide applications in machine learning such as $\ell_1$-regularized SVM [1], MAP inference [2], nonnegative matrix factorization [3], exemplar-based clustering [4, 5], sparse inverse covariance estimation [6], and Markov Decision Process [7]. However, as the demand for scalability keeps increasing, the scalability of existing LP solvers has become unsatisfactory. In particular, most algorithms in machine learning targeting large-scale data have a complexity linear to the data size [8, 9, 10], while the complexity of state-of-the-art LP solvers (i.e. Interior-Point method and Primal, Dual Simplex methods) is still at least quadratic in the number of variables or constraints [11].

The quadratic complexity comes from the need to solve each linear system exactly in both simplex and interior point method. In particular, the simplex method, when traversing from one corner point to another, requires solution to a linear system that has dimension linear to the number of variables or constraints, while in an Interior-Point method, finding the Newton direction requires solving a linear system of similar size. While there are sparse variants of *LU* and *Cholesky decomposition* that can utilize the sparsity pattern of matrix in a linear system, the worst-case complexity for solving such system is at least quadratic to the dimension except for very special cases such as a tri-diagonal or band-structured matrix.

For interior point method (IPM), one remedy to the high complexity is employing an iterative method such as Conjugate Gradient (CG) to solve each linear system inexactly. However, this can hardly tackle the ill-conditioned linear systems produced by IPM when iterates approach boundary of constraints [12]. Though substantial research has been devoted to the development of preconditioners that can help iterative methods to mitigate the effect of ill-conditioning [12, 13], creating a preconditioner of tractable size is a challenging problem by itself [13]. Most commercial LP software thus still relies on exact methods to solve the linear system.

On the other hand, some dual or primal (stochastic) sub-gradient descent methods have cheap cost for each iteration, but require $O(1/\epsilon^2)$ iterations to find a solution of $\epsilon$ precision, which in practice can even hardly find a feasible solution satisfying all constraints [14].

Augmented Lagrangian Method (ALM) was invented early in 1969, and since then there have been several works developed Linear Program solver based on ALM [15, 16, 17]. However, the challenge of ALM is that it produces a series of *bound-constrained quadratic problems* that, in the traditional sense, are harder to solve than linear system produced by IPM or Simplex methods [17]. Specifically, in a Projected-CG approach [18], one needs to solve several linear systems via CG to find solution to the bound-constrained quadratic program, while there is no guarantee on how many iterations it requires. On the other hand, Projected Gradient Method (PGM), despite its guaranteed iteration complexity, has very slow convergence in practice. More recently, Multi-block ADMM [19, 20] was proposed as a variant of ALM that, for each iteration, only updates one pass (or even less) blocks of primal variables before each dual update, which however, requires a much smaller step size in the dual update to ensure convergence [20, 21] and thus requires large number of iterations for convergence to moderate precision. To our knowledge, there is still no report on a significant improvement of ALM-based methods over IPM or Simplex method for Linear Programming.

In the recent years, Coordinate Descent (CD) method has demonstrated efficiency in many machine learning problems with bound constraints or other non-smooth terms [9, 10, 22, 23, 24, 25] and has solid analysis on its iteration complexity [26, 27]. In this work, we show that CD algorithm can be naturally combined with ALM to solve Linear Program more efficiently than existing methods on large-scale problems. We provide an $O((\log(1/\epsilon))^2)$ iteration complexity of the Augmented Lagrangian with Coordinate Descent (AL-CD) algorithm that bounds the total number of CD updates required for an $\epsilon$-precise solution, and describe an implementation of AL-CD that has cost $O(nnz(A))$ for each pass of CD. In practice, an active-set strategy is introduced to further reduce cost of each iteration to the active size of variables and constraints for *primal-sparse* and *dual-sparse* LP respectively, where a *primal-sparse* LP has most of variables being zero, and a *dual-sparse* LP has few binding constraints at the optimal solution. Note, unlike in IPM, the conditioning of each subproblem in ALM does not worsen over iterations [15, 16]. The AL-CD framework thus provides an alternative to interior point and simplex methods when it is infeasible to exactly solving an $n \times n$ (or $m \times m$) linear system.

## 2 Sparse Linear Program

We are interested in solving linear programs of the form

$$
\begin{aligned}
\min_{x \in \mathbb{R}^n} \quad & f(x) = c^T x \\
s.t. \quad & A_I x \le b_I \ , \ A_E x = b_E \\
& x_j \ge 0, \ \ j \in [n_b]
\end{aligned}
\tag{1}
$$

where $A_I$ is $m_I$ by $n$ matrix of coefficients and $A_E$ is $m_E$ by $n$. Without loss of generality, we assume non-negative constraints are imposed on the first $n_b$ variables, denoted as $x_b$, such that $x = [x_b; \ x_f]$ and $c = [c_b; \ c_f]$. The inequality and equality coefficient matrices can then be partitioned as $A_I = [A_{I,b} \ A_{I,f}]$ and $A_E = [A_{E,b} \ A_{E,f}]$. The dual problem of (1) then takes the form

$$
\begin{aligned}
\min_{y \in \mathbb{R}^m} \quad & g(y) = b^T y \\
s.t. \quad & -A_b^T y \le c_b \ , \ -A_f^T y = c_f \\
& y_i \ge 0, \ \ i \in [m_I].
\end{aligned}
\tag{2}
$$

where $m = m_I + m_E$, $b = [b_I; b_E]$, $A_b = [A_{I,b}; A_{E,b}]$, $A_f = [A_{I,f}; A_{E,f}]$, and $y = [y_I; y_E]$. In most of LP occur in machine learning, $m$ and $n$ are both at scale in the order $10^5 \text{~} 10^6$, for which an algorithm with cost $O(mn)$, $O(n^2)$ or $O(m^2)$ is unacceptable. Fortunately, there are usually various types of sparsity present in the problem that can be utilized to lower the complexity.

First, the constraint matrix $A = [A_I; A_E]$ are usually pretty sparse in the sense that $nnz(A) \ll mn$, and one can compute matrix-vector product $Ax$ in $O(nnz(A))$. However, in most of current LP solvers, not only matrix-vector product but also a linear system involving $A$ needs to be solved, which in general, has cost much more than $O(nnz(A))$ and can be up to $O(\min(n^3, m^3))$ in the worst case. In particular, the simplex-type methods, when moving from one corner to another, requires solving a linear system that involves a sub-matrix of $A$ with columns corresponding to the basic variables [11], while in an interior point method (IPM), one also needs to solve a *normal equation* system of matrix $AD_tA^T$ to obtain the Newton direction, where $D_t$ is a diagonal matrix that gradually enforces complementary slackness as IPM iteration $t$ grows [11]. While one remedy to the high complexity is to employ iterative method such as *Conjugate Gradient (CG)* to solve the system inexactly within IPM, this approach can hardly handle the ill-conditionedness occurs when IPM iterates approaches boundary [12]. On the other hand, the Augmented Lagrangian approach does not have such asymptotic ill-conditionedness and thus an iterative method with complexity linear to $O(nnz(A))$ can be used to produce sufficiently accurate solution for each sub-problem.

Besides sparsity in the constraint matrix $A$, two other types of structures, which we termed *primal* and *dual sparsity*, are also prevalent in the context of machine learning. A *primal-sparse* LP refers to an LP with optimal solution $x^*$ comprising only few non-zero elements, while a *dual-sparse* LP refers to an LP with few binding constraints at optimal, which corresponds to the non-zero dual variables. In the following, we give two examples of sparse LP.

**L1-Regularized Support Vector Machine**    The problem of L1-regularized multi-class Support Vector Machine [1]

$$\min_{w_m, \xi_i} \quad \lambda \sum_{m=1}^{k} \|w_m\|_1 + \sum_{i=1}^{l} \xi_i \tag{3}$$
$$s.t. \quad w_{y_i}^T x_i - w_m^T x_i \geq e_i^m - \xi_i, \ \forall (i, m)$$

where $e_i^m = 0$ if $y_i = m$, $e_i^m = 1$ otherwise. The task is dual-sparse since among all samples $i$ and class $k$, only those leads to misclassification will become binding constraints. The problem (3) is also primal-sparse since it does feature selection through $\ell_1$-penalty. Note the constraint matrix in (3) is also sparse since each constraint only involves two weight vectors, and the pattern $x_i$ can be also sparse.

**Sparse Inverse Covariance Estimation**    The Sparse Inverse Covariance Estimation aims to find a sparse matrix $\Omega$ that approximate the inverse of Covariance matrix. One of the most popular approach to this solves a program of the form [6]

$$\min_{\Omega \in \mathbb{R}^{d \times d}} \quad \|\Omega\|_1 \tag{4}$$
$$s.t. \quad \|S\Omega - I_d\|_{max} \leq \lambda$$

which is primal-sparse due to the $\|.\|_1$ penalty. The problem has a dense constraint matrix, which however, has special structure where the coefficient matrix $S$ can be decomposed into a product of two low-rank and (possibly) sparse $n$ by $d$ matrices $S = Z^T Z$. In case $Z$ is sparse or $n \ll d$, this decomposition can be utilized to solve the Linear Program much more efficiently. We will discuss on how to utilize such structure in section 4.3.

## 3  Primal and Dual Augmented Coordinate Descent

In this section, we describe an Augmented Lagrangian method (ALM) that carefully tackles the sparsity in a LP. The choice between Primal and Dual ALM depends on the type of sparsity present in the LP. In particular, a primal AL method can solve a problem of few non-zero variables more efficiently, while dual ALM will be more efficient for problem with few binding constraints. In the following, we describe the algorithm only from the primal point of view, while the dual version can be obtained by exchanging the roles of primal (1) and dual (2).

**Algorithm 1** (Primal) Augmented Lagrangian Method

---

Initialization: $y^0 \in \mathbb{R}^m$ and $\eta_0 > 0$.

**repeat**

    1. Solve (6) to obtain $(x^{t+1}, \xi^{t+1})$ from $y^t$.

    2. Update $y^{t+1} = y^t + \eta_t \begin{bmatrix} A_I x^{t+1} - b_I + \xi^{t+1} \\ A_E x^{t+1} - b_E \end{bmatrix}$.

    3. $t = t + 1$.

    4. Increase $\eta_t$ by a constant factor if necessary.

**until** $\|[A_I x^t - b_I]_+\|_\infty \le \epsilon_p$ and $\|A_E x^t - b_E\|_\infty \le \epsilon$.

---

### 3.1 Augmented Lagrangian Method (Dual Proximal Method)

Let $g(y)$ be the dual objective function (2) that takes $\infty$ if $y$ is infeasible. The primal AL algorithm can be interpreted as a *dual proximal point* algorithm [16] that for each iteration $t$ solves

$$y^{t+1} = \underset{y}{argmin} \quad g(y) + \frac{1}{2\eta_t}\|y - y^t\|^2. \tag{5}$$

Since $g(y)$ is nonsmooth, (5) is not easier to solve than the original dual problem. However, the dual of (5) takes the form:

$$\min_{x,\,\xi} \quad F(x,\xi) = c^T x + \frac{\eta_t}{2}\left\| \begin{bmatrix} A_I x - b_I + \xi \\ A_E x - b_E \end{bmatrix} + \frac{1}{\eta_t}\begin{bmatrix} y_I^t \\ y_E^t \end{bmatrix} \right\|^2 \tag{6}$$
$$s.t. \quad x_b \ge 0,\ \xi \ge 0,$$

which is a bound-constrained quadratic problem. Note given $(x, \xi)$ as Lagrangian Multipliers of (5), the corresponding $y$ minimizing Lagrangian $\mathcal{L}(x, \xi, y)$ is

$$y(x,\xi) = \eta_t \begin{bmatrix} A_I x - b_I + \xi \\ A_E x - b_E \end{bmatrix} + \begin{bmatrix} y_I^t \\ y_E^t \end{bmatrix}, \tag{7}$$

and thus one can solve $(x^*, \xi^*)$ from (6) and find $y^{t+1}$ through (7). The resulting algorithm is sketched in Algorithm 1. For problem of medium scale, (6) is not easier to solve than a linear system due to non-negative constraints, and thus an ALM is not preferred to IPM in the traditional sense. However, for large-scale problem with $m \times n \gg nnz(A)$, the ALM becomes advantageous since: (i) the conditioning of (6) does not worsen over iterations, and thus allows iterative methods to solve it approximately in time proportional to $O(nnz(A))$. (ii) For a primal-sparse (dual-sparse) problem, most of primal (dual) variables become binding at zero as iterates approach to the optimal solution, which yields a potentially much smaller subproblem.

### 3.2 Solving Subproblem via Coordinate Descent

Given a dual solution $y_t$, we employ a variant of Randomized Coordinate Descent (RCD) method to solve subproblem (6). First, we note that, given $x$, the part of variables in $\xi$ can minimized in closed-form as

$$\xi(x) = [b_I - A_I x - y_I^t / \eta_t]_+, \tag{8}$$

where function $[v]_+$ truncates each element of vector $v$ to be non-negative as $[v]_{+i} = \max\{v_i, 0\}$. Then (6) can be re-written as

$$\min_x \quad \hat{F}(x) = c^T x + \frac{\eta_t}{2}\left\| \begin{bmatrix} [A_I x - b_I + y_I^t/\eta_t]_+ \\ A_E x - b_E + y_E^t/\eta_t \end{bmatrix} \right\|^2 \tag{9}$$
$$s.t. \quad x_b \ge 0.$$

| **Algorithm 2** RCD for subproblem (6) | **Algorithm 3** PN-CG for subproblem (6) |
|---|---|
| INPUT: $\eta_t > 0$ and $(x^{t,0}, w^{t,0}, v^{t,0})$ satisfying relation (11), (12). | INPUT: $\eta_t > 0$ and $(x^{t,0}, w^{t,0}, v^{t,0})$ satisfying relation (11), (12). |
| OUTPUT: $(x^{t,k}, w^{t,k}, v^{t,k})$ | OUTPUT: $(x^{t,k}, w^{t,k}, v^{t,k})$ |
| **repeat** | **repeat** |
|   1. Pick a coordinate $j$ uniformly at random. |   1. Identify active variables $\mathcal{A}^{t,k}$. |
|   2. Compute $\nabla_j \hat{F}(x)$, $\nabla_j^2 \hat{F}(x)$. |   2. Compute $[\nabla_j F(x)]_{\mathcal{A}^{t,k}}$ and set $\mathcal{D}^{t,k}$. |
|   3. Obtain Newton direction $d_j^*$. |   3. Find Newton direction $d_{\mathcal{A}^{t,k}}^*$ with CG. |
|   4. Do line search (15) to find step size. |   4. Find step size via projected line search. |
|   5. Update $x^{t,k+1} \leftarrow x^{t,k} + \beta^r d_j^*$. |   5. Update $x^{t,k+1} \leftarrow (x^{t,k} + \beta^r d_j^*)_+$. |
|   6. Maintain relation (11), (12). |   6. Maintain relation (11), (12). |
|   7. $k \leftarrow k+1$. |   7. $k \leftarrow k+1$. |
| **until** $\|d^*(x)\|_\infty \le \epsilon_t$. | **until** $\|d_{\mathcal{A}^{t,k}}^*\|_\infty \le \epsilon_t$. |

Denote the objective function as $\hat{F}(x)$. The gradient of (9) can be expressed as

$$\nabla \hat{F}(x) = c + \eta_t A_I^T [w]_+ + \eta_t A_E^T v \tag{10}$$

where

$$w = A_I x - b_I + y_I^t / \eta_t \tag{11}$$

$$v = A_E x - b_E + y_E^t / \eta_t, \tag{12}$$

and the (generalized) Hessian of (9) is

$$\nabla^2 \hat{F}(x) = \eta_t A_I^T D(w) A_I + \eta_t A_E^T A_E, \tag{13}$$

where $D(w)$ is an $m_I$ by $m_I$ diagonal matrix with $D_{ii}(w) = 1$ if $w_i > 0$ and $D_{ii} = 0$ otherwise.

The RCD algorithm then proceeds as follows. In each iteration $k$, it picks a coordinate from $j \in \{1, .., n\}$ uniformly at random and minimizes w.r.t. the coordinate. The minimization is conducted by a single-variable Newton step, which first finds the Newton direction $d_j^*$ through minimizing a quadratic approximation

$$\begin{aligned} d_j^* = \underset{d}{argmin} \quad & \nabla_j \hat{F}(x^{t,k}) d + \frac{1}{2} \nabla_j^2 \hat{F}(x^{t,k}) d^2 \\ s.t. \quad & x_j^{t,k} + d \ge 0, \end{aligned} \tag{14}$$

and then conducted a line search to find the smallest $r \in \{0, 1, 2, ...\}$ satisfying

$$\hat{F}(x^{t,k} + \beta^r d_j^* e_j) - \hat{F}(x^{t,k}) \le \sigma \beta^r (\nabla_j \hat{F}(x^{t,k}) d_j^*). \tag{15}$$

for some line-search parameter $\sigma \in (0, 1/2]$, $\beta \in (0, 1)$, where $e_j$ denotes a vector with only $j$th element equal to 1 and all others equal to 0. Note the single-variable problem (14) has closed-form solution

$$d_j^* = \left[ x_j^{t,k} - \nabla_j \hat{F}(x_j^{t,k}) / \nabla_j^2 \hat{F}(x_j^{t,k}) \right]_+ - x_j^{t,k}, \tag{16}$$

which in a naive implementation, takes $O(nnz(A))$ time due to the computation of (11) and (12). However, in a clever implementation, one can maintain the relation (11), (12) as follows whenever a coordinate $x_j$ is updated by $\beta^r d_j^*$

$$\begin{bmatrix} w^{t,k+1} \\ v^{t,k+1} \end{bmatrix} = \begin{bmatrix} w^{t,k} \\ v^{t,k} \end{bmatrix} + \beta^r d_j^* \begin{bmatrix} a_j^I \\ a_j^E \end{bmatrix}, \tag{17}$$

where $a_j = [a_j^I; \ a_j^E]$ denotes the $j$th column of $A_I$ and $A_E$. Then the gradient and (generalized) second-derivative of $j$th coordinate

$$\nabla_j \hat{F}(x) = c_j + \eta_t \langle a_j^I, [w]_+ \rangle + \eta_t \langle a_j^E, v \rangle$$

$$\nabla_j^2 \hat{F}(x) = \eta_t \left( \sum_{i: w_i > 0} (a_{i,j}^I)^2 + \sum_i (a_{i,j}^E)^2 \right) \tag{18}$$

can be computed in $O(nnz(a_j))$ time. Similarly, for each coordinate update, one can evaluate the difference of function value $\hat{F}(x^{t,k} + d_j^* e_j) - \hat{F}(x^{t,k})$ in $O(nnz(a_j))$ by only computing terms related to the $j$th variable.

The overall procedure for solving subproblem is summarized in Algorithm 2. In practice, a random permutation is used instead of uniform sampling to ensure that every coordinate is updated once before proceeding to the next round, which can speed up convergence and ease the checking of stopping condition $\|d^*(x)\|_\infty \leq \epsilon_t$, and an active-set strategy is employed to avoid updating variables with $d_j^* = 0$. We describe details in section 4

### 3.3 Convergence Analysis

In this section, we prove the iteration complexity of AL-CD method. Existing analysis [26, 27] shows that Randomized Coordinate Descent can be up to $n$ times faster than Gradient-based methods in certain conditions. However, to prove a global linear rate of convergence the analysis requires objective function to be strongly convex, which is not true for our sub-problem (6). Here we follow the approach in [28, 29] to show global linear convergence of Algorithm 2 by utilizing the fact that, when restricted to a constant subspace, (6) is strongly convex. All proofs will be included in the appendix.

**Theorem 1** (Linear Convergence). *Denote $F^*$ as the optimum of (6) and $\bar{x} = [x; \xi]$. The iterates $\{\bar{x}^k\}_{k=0}^\infty$ of the RCD Algorithm 2 has*

$$\mathbb{E}[F(\bar{x}^{k+1})] - F^* \leq \left(1 - \frac{1}{\gamma n}\right) \left(\mathbb{E}[F(\bar{x}^k)] - F^*\right), \tag{19}$$

*where*

$$\gamma = \max\left\{16\eta_t M\theta(F^0 - F^*),\ 2M\theta(1 + 4L_g^2),\ 6\right\},$$

*$M = \max_{j \in [\bar{n}]} \|\bar{a}_j\|^2$ is an upper bound on coordinate-wise second derivative, and $L_g$ is local Lipschitz-continuous constant of function $g(z) = \eta_t \|z - b + y_t/\eta_t\|^2$, and $\theta$ is constant of Hoffman's bound that depends on the polyhedron formed by the set of optimal solutions.*

Then the following theorem gives a bound on the number of iterations required to find an $\epsilon_0$-precise solution in terms of the proximal minimization (5).

**Theorem 2** (Inner Iteration Complexity). *Denote $y(\bar{x}^k)$ as the dual solution (7) corresponding to the primal iterate $\bar{x}^k$. To guarantee*

$$\|y(\bar{x}^k) - y^{t+1}\| \leq \epsilon_0 \tag{20}$$

*with probability $1 - p$, it suffices running RCD Algorithm 2 for number of iterations*

$$k \geq 2\gamma n \log \left(\sqrt{\frac{2(F(\bar{x}^0) - F^*)}{\eta_t p}} \frac{1}{\epsilon_0}\right).$$

Now we prove the overall iteration complexity of AL-CD. Note that existing linear convergence analysis of ALM on Linear Program [16] assumes exact solutions of subproblem (6), which is not possible in practice. Our next theorem extends the linear convergence result to cases when subproblems are solved *inexactly*, and in particular, shows the total number of coordinate descent updates required to find an $\epsilon$-accurate solution.

**Theorem 3** (Iteration Complexity). *Denote $\{\hat{y}^t\}_{t=1}^\infty$ as the sequence of iterates obtained from inexact dual proximal updates, $\{y^t\}_{t=1}^\infty$ as that generated by exact updates, and $y_{S^*}$ as the projection of $y$ to the set of optimal dual solutions. To guarantee $\|\hat{y}^t - \hat{y}_{S_*}^t\| \leq 2\epsilon$ with probability $1 - p$, it suffices to run Algorithm 1 for*

$$T = (1 + \frac{1}{\alpha}) \log\left(\frac{LR}{\epsilon}\right) \tag{21}$$

*outer iterations with $\eta_t = (1 + \alpha)L$, and solve each sub-problem (6) by running Algorithm 2 for*

$$k \geq 2\gamma n \left(\log\left(\frac{\omega}{\epsilon}\right) + \frac{3}{2} \log\left((1 + \frac{1}{\alpha}) \log\frac{LR}{\epsilon}\right)\right) \tag{22}$$

*inner iterations, where $L$ is a constant depending on the polyhedral set of optimal solutions, $\omega = \sqrt{\frac{2(1+\alpha)L(F^0 - F^*)}{p}}$, $R = \|\mathbf{prox}_{\eta_t g}(y^0) - y^0\|$, and $F^0$, $F^*$ are upper and lower bounds on the initial and optimal function values of subproblem respectively.*

### 3.4 Fast Asymptotic Convergence via Projected Newton-CG

The RCD algorithm converges to a solution of moderate precision efficiently, but in some problems a higher precision might be required. In such case, we transfer the subproblem solver from RCD to a *Projected Newton-CG (PN-CG)* method after iterates are close enough to the optimum. Note the Projected Newton method does not have global iteration complexity but has fast convergence for iterates very close to the optimal.

Denote $F(x)$ as the objective in (9). Each iterate of PN-CG begins by finding the set of *active variables* defined as

$$\mathcal{A}^{t,k} = \{j | x_j^{t,k} > 0 \vee \nabla_j F(x^{t,k}) < 0\}. \tag{23}$$

Then the algorithm fixes $x_j^{t,k} = 0, \forall j \notin A^{t,k}$ and solves a Newton linear system w.r.t. $j \in \mathcal{A}^{t,k}$

$$[\nabla^2_{\mathcal{A}^{t,k}} F(x^{t,k})]d = -[\nabla_{\mathcal{A}^{t,k}} F(x^{t,k})] \tag{24}$$

to obtain direction $d^*$ for the current active variables. Let $d_{A^{t,k}}$ denotes a size-$n$ vector taking value in $d^*$ for $j \in A^{t,k}$ and taking value 0 for $j \notin A^{t,k}$. The algorithm then conducts a *projected line search* to find smallest $r \in \{0, 1, 2, ...\}$ satisfying

$$F([x^{t,k} + \beta^r d_{A^{t,k}}]_+) - F(x^{t,k}) \leq \sigma\beta^r (\nabla_j F(x^{t,k}) d_{A^{t,k}}), \tag{25}$$

and update $x$ by $x^{t,k+1} \leftarrow (x^{t,k} + \beta^r d_j^*)_+$. Compared to interior point method, one key to the tractability of this approach lies on the conditioning of linear system (24), which does not worsen as outer iteration $t$ increases, so an iterative *Conjugate Gradient (CG)* method can be used to obtain accurate solution without factorizing the Hessian matrix. The only operation required within CG is the Hessian-vector product

$$[\nabla^2_{\mathcal{A}^{t,k}} F(x^{t,k})]s = \eta_t \, [A_I^T D(w^{t,k}) A_I + A_E^T A_E]_{\mathcal{A}^{t,k}} \, s, \tag{26}$$

where the operator $[.]_{\mathcal{A}^{t,k}}$ takes the sub-matrix with row and column indices belonging to $\mathcal{A}^{t,k}$. For a *primal or dual-sparse* LP, the product (26) can be evaluated very efficiently, since it only involves non-zero elements in columns of $A_I$, $A_E$ belonging to the active set, and rows of $A_I$ corresponding to the binding constraints for which $D_{ii}(w^{t,k}) > 0$. The overall cost of the product (26) is only

$$O\left(nnz([A_I]_{\mathcal{D}^{t,k}, \mathcal{A}^{t,k}}) + nnz([A_E]_{:, \mathcal{A}^{t,k}})\right),$$

where $\mathcal{D}^{t,k} = \{i | w_i^{t,k} > 0\}$ is the set of current binding constraints. Considering that the computational bottleneck of PN-CG is on the *CG* iterations for solving linear system (24), the efficient computation of product (26) reduces the overall complexity of PN-CG significantly. The whole procedure is summarized in Algorithm 3.

## 4 Practical Issues

### 4.1 Precision of Subproblem Minimization

In practice, it is unnecessary to solve subproblem (6) to high precision, especially for iterations of ALM in the beginning. In our implementation, we employ a two-phase strategy, where in the first phase we limit the cost spent on each sub-problem (6) to be a constant multiple of $nnz(A)$, while in the second phase we dynamically increment the AL parameter $\eta_t$ and inner precision $\epsilon_t$ to ensure sufficient decrease in the primal and dual infeasibility respectively. The two-phase strategy is particularly useful for primal or dual-sparse problem, where sub-problem in the latter phase has smaller active set that results in less computation cost even when solved to high precision.

### 4.2 Active-Set Strategy

Our implementation of Algorithm 2 maintains an active set of variables $\mathcal{A}$, which initially contains all variables, but during the RCD iterates, any variable $x_j$ binding at 0 with gradient $\nabla_j F$ greater than a threshold $\delta$ will be excluded from $\mathcal{A}$ till the end of each subproblem solving. $\mathcal{A}$ will be re-initialized after each dual proximal update (7). Note in the initial phase, the cost spent on each subproblem is a constant multiple of $nnz(A)$, so if $|\mathcal{A}|$ is small one would spend more iterations on the active variables to achieve faster convergence.

### 4.3 Dealing with Decomposable Constraint Matrix

When we have a $m$ by $n$ constraint matrix $A = UV^T$ that can be decomposed into product of an $m \times r$ matrix $U$ and a $r \times n$ matrix $V^T$, if $r \ll \min\{m, n\}$ or $nnz(U) + nnz(V) \ll nnz(A)$, we can re-formulate the constraint $Ax \le b$ as $Uz \le b$ , $V^Tx = z$ with auxiliary variables $z \in \mathbb{R}^r$. This new representation reduce the cost of Hessian-vector product in Algorithm 3 and the cost of each pass of CD in Algorithm 2 from $O(nnz(A))$ to $O(nnz(U) + nnz(V))$.

## 5 Numerical Experiments

Table 1: Timing Results (in sec. unless specified o.w.) on Multiclass L1-regularized SVM

| Data | $n_b$ | $m_I$ | P-Simp. | D-Simp. | Barrier | D-ALCD | P-ALCD |
|---|---|---|---|---|---|---|---|
| rcv1 | 4,833,738 | 778,200 | > 48hr | > 48hr | > 48hr | 3,452 | **3,155** |
| news | 2,498,415 | 302,765 | > 48hr | 37,912 | > 48hr | **148** | 395 |
| sector | 11,597,992 | 666,848 | > 48hr | 9,282 | > 48hr | **1,419** | 2,029 |
| mnist | 75,620 | 540,000 | 6,454 | 2,556 | 73,036 | **146** | 7,207 |
| cod-rna.rf | 69,537 | 59,535 | 86,130 | 5,738 | > 48hr | 3,130 | **2,676** |
| vehicle | 79,429 | 157,646 | 3,296 | 143.33 | 8,858 | **31** | 598 |
| real-sim | 114,227 | 72,309 | > 48hr | 49,405 | 89,476 | **179** | 297 |

Table 2: Timing Results (in sec. unless specified o.w.) on Sparse Inverse Covariance Estimation

| Data | $n_b$ | $m_I$ | $m_E$ | $n_f$ | P-Simp | D-Simp | Barrier | D-ALCD | P-ALCD |
|---|---|---|---|---|---|---|---|---|---|
| textmine | 60,876 | 60,876 | 43,038 | 43,038 | > 48hr | > 48hr | > 48hr | 43,096 | **18,507** |
| E2006 | 55,834 | 55,834 | 32,174 | 32,174 | > 48hr | > 48hr | 94623 | > 48hr | **4,207** |
| dorothea | 47,232 | 47,232 | 1,600 | 1,600 | 3,980 | 103 | 82 | 47 | **38** |

Table 3: Timing Results (in sec. unless specified o.w.) for Nonnegative Matrix Factorization.

| Data | $n_b$ | $m_I$ | P-Simp. | D-Simp. | Barrier | D-ALCD | P-ALCD |
|---|---|---|---|---|---|---|---|
| micromass | 2,896,770 | 4,107,438 | > 96hr | > 96hr | 280,230 | 12,966 | **12,119** |
| ocr | 6,639,433 | 13,262,864 | > 96hr | > 96hr | 284,530 | **40,242** | > 96hr |

In this section, we compare the AL-CD algorithm with state-of-the-art implementation of interior point and primal, dual Simplex methods in commercial LP solver CPLEX, which is of top efficiency among many LP solvers as investigated in [30]. For all experiments, the stopping criteria is set to require both primal and dual infeasibility (in the $\ell_\infty$-norm) smaller than $10^{-3}$ and set the initial sub-problem tolerance $\epsilon_t = 10^{-2}$ and $\eta_t = 1$. The LP instances are generated from L1-SVM (3), Sparse Inverse Covariance Estimation (4) and Nonnegative Matrix Factorization [3]. For the Sparse Inverse Covariance Estimation problem, we use technique introduced in section 4.3 to decompose the low-rank matrix $S$, and since (4) results in $d$ independent problems for each column of the estimated matrix, we report result on only one of them. The data source and statistics are included in the appendix.

Among all experiments, we observe that the proposed primal, dual AL-CD methods become particularly advantageous when the matrix $A$ is sparse. For example, for text data set *rcv1*, *real-sim* and *news* in Table 1, the matrix $A$ is particularly sparse and AL-CD can be orders of magnitude faster than other approaches by avoiding solving $n \times n$ linear system exactly. In addition, the dual-ALCD (also dual simplex) is more efficient in L1-SVM problem due to the problem's strong dual sparsity, while the primal-ALCD is more efficient on the primal-sparse Inverse Covariance estimation problem. For the Nonnegative Matrix Factorization problem, both the dual and primal LP solutions are not particularly sparse due to the choice of matrix approximation tolerance (1% of #samples), but the AL-CD approach is still comparably more efficient.

**Acknowledgement** We acknowledge the support of ARO via W911NF-12-1-0390, and the support of NSF via grants CCF-1320746, CCF-1117055, IIS-1149803, IIS-1320894, IIS-1447574, DMS-1264033, and NIH via R01 GM117594-01 as part of the Joint DMS/NIGMS Initiative to Support Research at the Interface of the Biological and Mathematical Sciences.

## Footnotes

[1] Our solver has been released here: http://www.cs.utexas.edu/~ianyen/LPsparse/

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
