[Supplementary Material]

# A   Appendix-A — Proof for Convergence Analysis

## A.1   Linear Convergence of Augmented Lagrangian Method

**Theorem 4.** *Let $\{y^t\}_{t=0}^{\infty}$ be the sequences of dual variables produced by Algorithm 1 and $\{(x^t, \xi_t)\}_{t=0}^{\infty}$ be the corresponding sequence of solutions to the primal Augmented Lagrangian problem. Denote*

$$\Delta^t = \frac{1}{\eta_t}(y^{t+1} - y^t) = \begin{bmatrix} A_I x^t - b_I + \xi^t \\ A_E x^t - b_E \end{bmatrix} \in \partial g(y^t). \tag{27}$$

*and $\Pi_{S^*}(y^t)$ as the projection of $y^t$ to the set of optimal dual solutions. Then we have*

$$\|y^t - \Pi_{S^*}(y^t)\| \leq L\|\Delta^t\| \tag{28}$$

*and*

$$\|\Delta^{t+1}\| \leq \min\left(\frac{L}{\eta_t}, 1\right)\|\Delta^t\|, \tag{29}$$

*where $L := L(S^*, y^0) > 0$ is a constant depending on the solution set $S^*$ and initial distance to this set $R = \|y^0 - \Pi_{S^*}(y^0)\|$.*

*Proof.* This theorem is a special case of the linear convergence proof in [16]. In particular, the Linear Program (1) can be written as

$$\begin{aligned}
\min_{x \in \mathbb{R}^n} \quad & f(x) = c^T x \\
s.t. \quad & \begin{bmatrix} A_I & I \\ A_E & O \end{bmatrix}\begin{bmatrix} x \\ \xi \end{bmatrix} = \begin{bmatrix} b_I \\ b_E \end{bmatrix}, \\
& x_j \geq 0, \quad j = 1...n_b \\
& \xi_i \geq 0, \quad i = 1...m_I,
\end{aligned} \tag{30}$$

which is a special case of the Quadratic Programming formulation analyzed in [16] with quadratic term $Q = 0$ (which is positive semi-definite). The analysis assumes all iterates $y^t$ to be within a bounded distance $R$ to the optimal solution set, which is satisfied with $R = \|y^0 - \Pi_{S^*}(y^t)\|$ since by non-expansiveness of proximal operator, we have

$$\|y^{t+1} - \Pi_{S^*}(y^{t+1})\| \leq \|y^{t+1} - \Pi_{S^*}(y^t)\| = \|\mathbf{prox}_g(y^t) - \mathbf{prox}_g(\Pi_{S^*}(y^t))\| \leq \|y^t - \Pi_{S^*}(y^t)\|,$$

where

$$\mathbf{prox}_g(y^t) = \underset{y}{argmin}\ g(y) + \frac{\eta_t}{2}\|y - y^t\|^2,$$

and thus the distance of each iterate to the optimal set is bounded by $R = \|y^0 - \Pi_{S^*}(y^0)\|$. Inequalities (28), (29) then follow from Proposition 4.4 and Theorem 4.5 of [16] respectively, where the constant $L$ is defined through characteristics of $S^*$ and an upper bound $R$ on the distance to solution set. □

We then have following outer iteration complexity for Algorithm 1, assuming each proximal sub-problem (6) is solved exactly.

**Corollary 1** (Outer Iteration Complexity). *Setting $\eta_t \geq \eta = (1 + \alpha)L$, we have*

$$\|y^t - \Pi_{S^*}(y^t)\| \leq \epsilon$$

*by performing*

$$t \geq (1 + \frac{1}{\alpha})\log\left(\frac{L\|\Delta^0\|}{\epsilon}\right)$$

*iterations of Algorithm (1), where $\|\Delta^0\| = \|\mathbf{prox}_{\eta_t g}(y^0) - y^0\|$.*

*Proof.* For $\eta_t \geq \eta = (1 + \alpha)L$, we have

$$\|\Delta^{t+1}\| \leq (1 - \frac{1}{z})\|\Delta^t\|,$$

where $z = (1 + \frac{1}{\alpha})$, and thus for

$$t \geq (1 + \frac{1}{\alpha}) \log \left( \frac{L\|\Delta^0\|}{\epsilon} \right),$$

we have

$$\|\Delta^t\| \leq (1 - \frac{1}{z})^{z \log \frac{L\|\Delta^0\|}{\epsilon}} \|\Delta^0\| \leq (e^{-1})^{\log \frac{L\|\Delta^0\|}{\epsilon}} \|\Delta_0\| \leq \frac{\epsilon}{L},$$

and therefore by (28), $\|y^t - \Pi_{S^*}(y^t)\| \leq \epsilon$. $\qquad\square$

## A.2 Linear Convergence of Randomized Coordinate Descent on Subproblem (6)

In this section, we prove linear convergence of Algorithm 2 to the optimum of sub-problem (6) by exploiting the fact that objective (6), though not being strongly convex, has strong convexity when restricted to a constant linear subspace [28, 29]. In particular, denote $\bar{n} = n + m_I$ and

$$\bar{x} = \begin{bmatrix} x \\ \xi \end{bmatrix} \in \mathbb{R}^{\bar{n}} , \; \bar{c} = \begin{bmatrix} c \\ 0 \end{bmatrix} , \; \bar{A} = \begin{bmatrix} A_I & I \\ A_E & O \end{bmatrix}.$$

We can express the objective (6) as

$$\min_{x, x_b \geq 0, \xi \geq 0} \quad F(\bar{x}) = \bar{c}^T \bar{x} + g(\bar{A}\bar{x}), \tag{31}$$

where

$$g(z) = \frac{\eta_t}{2} \|z - b - \frac{1}{\eta_t} y^t\|^2$$

is $\eta_t$-strongly convex w.r.t. $z$ and therefore $F(\bar{x})$ is strongly convex when restricted to the space $\mathcal{N}^\perp$, where $\mathcal{N} = Null(\bar{A})$ is the Nullspace of constraint matrix $\bar{A}$. Formally, a Constant Nullspace Strongly Convex (CNSC) function has the following properties.

**Lemma 1** ( CNSC [29] ). *Let* $\mathcal{N} = Null(\bar{A})$ *be the Nullspace of* $\bar{A}$ *and* $H = \nabla^2 F(\bar{x})$ *be the Hessian matrix of* (31). *For any* $\bar{x} \in \mathbb{R}^{\bar{n}}$, *we can express it as* $\bar{x} = u + v$ *where* $u = \Pi_{\mathcal{N}}(\bar{x})$, $v = \Pi_{\mathcal{N}^\perp}(\bar{x})$ *s.t.*

$$Hu = 0 \tag{32}$$

*and*

$$v^T H v \geq m\|v\|^2, \tag{33}$$

*for some* $m > 0$.

*Proof.* The Hessian of (31) can be written as

$$\nabla^2 F(\bar{x}) = H = \eta_t \bar{A}^T \bar{A}$$

and thus (32) can be easily verified. On the other hand, (33) holds with $m = \eta_t \lambda_{min} > 0$, where $\lambda_{min}$ denotes minimum *positive* eigenvalue of $\bar{A}^T \bar{A}$. $\qquad\square$

Then we can profile the optimal solution of (31) with the following condition.

**Lemma 2** (Optimality Condition). *Express subproblem objective* (31) *as*

$$F(\bar{x}) + h(\bar{x}),$$

*where* $h(\bar{x}) = \sum_{j \in [\bar{n}] \setminus \{n_b+1...n_b+n_f\}} h_j(\bar{x}_j)$ *with*

$$h_j(\bar{x}) = \begin{cases} 0 & , \bar{x}_j \geq 0 \\ \infty & , o.w.. \end{cases} \tag{34}$$

*Then there are unique* $\rho^*$, $s^*$ *and* $t^*$ *s.t.* $\bar{x}^*$ *is optimal solution of* (31) *iff*

$$-\nabla F(\bar{x}^*) = -\bar{c} - \nabla g(t^*) = \rho^* \in \partial h(\bar{x}) \tag{35}$$

*and* $\bar{c}^T \bar{x}^* = s^*$ *and* $\bar{A}\bar{x}^* = t^*$.

*Proof.* The first-order condition implies (35) to be necessary and sufficient for $\bar{x}^*$ to be optimal, so we only need to verify the uniqueness of $\rho^*$, $s^*$ and $t^*$. Consider two solutions $\bar{x}_1$, $\bar{x}_2$ that are both optimal. Denote $\Delta x = \bar{x}_1 - \bar{x}_2$. By convexity of $h(\bar{x})$, we have

$$\langle -\nabla F(\bar{x}_1) + \nabla F(\bar{x}_2), \Delta x \rangle = \langle \rho_1 - \rho_2, \Delta x \rangle \geq 0, \tag{36}$$

Note for quadratic $F(\bar{x})$, the Hessian $\nabla^2 F(\bar{x}) = H$ is constant and thus

$$\nabla F(\bar{x}_1) - \nabla F(\bar{x}_2) = H \Delta x. \tag{37}$$

Then by CNSC condition (32), we have

$$\langle -\nabla F(\bar{x}_1) + \nabla F(\bar{x}_2), \Delta x \rangle = \langle -H\Delta x, \Delta x \rangle = -\Delta v^T H \Delta v \tag{38}$$

where $\Delta v = \Pi_{\mathcal{N}^\perp}(\Delta x)$ is the projection of $\Delta x$ onto the subspace $\mathcal{N}^\perp$. Then by CNSC (33),

$$-\Delta v^T H \Delta v \leq -m\|\Delta v\|^2$$

for some $m > 0$, but (36) implies

$$-\Delta v^T H \Delta v \geq 0.$$

Then the above two inequalities can simultaneously hold only if $\Delta v = 0$, which means the optimal $v^*$ as well as $t^* = \bar{A}\bar{x}^* = \bar{A}v^*$ are unique. Furthermore, the optimal $\rho^* = -\bar{c} - \nabla g(t^*)$ and $s^* = F^* - g(t^*)$ are also unique. $\square$

From Lemma 2, the set of optimal solutions forms a polyhedral set satisfying (i) $\bar{A}\bar{x} = t^*$, (ii) $\bar{c}^T \bar{x} = s^*$ and (iii) $x_b \geq 0$, $\xi \geq 0$. Then we can bound the distance of any point $\bar{x}$ to the polyhedral set by the amount of infeasibility to the three (in)equalities based on Hoffman's bound introduced as follows.

**Lemma 3** (Hoffman's Bound). *Let $\mathcal{S} = \{x \in \mathbb{R}^d \mid Ax \leq b, \ Ex = c\}$ be a polyhedral set. Then for any point $x \in \mathbb{R}^d$,*

$$\|x - \Pi_{\mathcal{S}}(x)\|_2^2 \leq \theta \left\| \begin{matrix} [Ax - b]_+ \\ Ex - c \end{matrix} \right\|_2^2 \tag{39}$$

*where $\Pi_{\mathcal{S}}(x) = \arg\min_{y \in \mathcal{S}} \|y - x\|$ is the projection of $x$ to the set $\mathcal{S}$, and $\theta > 0$ is a constant depending on the polyhedral set $\mathcal{S}$.*

*Proof.* The Hoffman's bound first appears in [31] and a proof for the $\ell_2$-norm's version (39) and the definition of the constant $\theta(\mathcal{S})$ can be found in [28] (lemma 4.3). $\square$

Note for any feasible descent method (such as Coordinate Descent method), all iterates $\{\bar{x}^k\}_{k=1}^\infty$ are feasible, and therefore one can bound the distance of any iterate to the set of optimal solutions by the amount of infeasibility to the two conditions $\bar{A}\bar{x} = t^*$, $\bar{c}^T \bar{x} = s^*$ as

$$\|\bar{x} - \Pi_{\mathcal{S}}(\bar{x})\|^2 \leq \theta(\mathcal{S}) \left( \|\bar{A}\bar{x} - t^*\|^2 + \|\bar{c}^T \bar{x} - s^*\|^2 \right), \tag{40}$$

which plays an important role in the proof of linear convergence of Randomized Coordinate Descent on the CNSC function (31). Now we move on to lemmas specific to Algorithm 2. For simplicity, we will analyze RCD that employs a conservative step size $1/\nabla_{jj}^2 \bar{F}(\bar{x}) = 1/(\eta_t \|\bar{a}_j\|^2)$ instead of the one using dynamic line search (15). However, the result only differs by a constant factor $\sigma\beta$ (line search parameter) on the descent amount.

**Lemma 4** (Descent Amount). *The expected descent amount for each RCD update of Algorithm 2 has*

$$\mathbb{E}[F(\bar{x}^{k+1})] - F(\bar{x}^k) \leq \frac{1}{n} \left( \min_\delta \ h(\bar{x}^k + \delta) + \langle \nabla F(\bar{x}^k), \delta \rangle + \frac{M\eta_t}{2}\|\delta\|^2 \right), \tag{41}$$

*where $M \geq \max_{j \in [n]} \|\bar{a}_j\|^2$ is an upper bound on the coordinate-wise second derivative, $\bar{a}_j$ is the $j$-th column of $\bar{A}$.*

*Proof.* First, notice that Algorithm 2 maintains $\xi$ (i.e. $\bar{x}_{n+1},..,\bar{x}_{n+m_f}$) to be optimal given other variables $x$ through equation (8), so we have

$$0 = \min_{\delta_j} \ h_j(\bar{x}_j + \delta_j) + \nabla_j F(\bar{x}^k)\delta_j + \frac{M\eta_t}{2}\delta_j^2 \ , \ j = n+1, ..., n+m_I. \qquad (42)$$

Therefore, the algorithm picks coordinate uniformly from $\{1...n\}$ (without $\{n+1,...,n+m_I\}$) to update. Note the constant

$$M \geq \frac{1}{\eta_t} \max_{j\in[n]} |\nabla_{jj}^2 F(\bar{x})| = \max_{j\in[\bar{n}]} \|\bar{a}_j\|^2$$

upper bounds the coordinate-wise second-derivative of both $F(\bar{x})$ and $\hat{F}(x) = \min_\xi F(x, \xi)$. Therefore, denote $e_j$ as vector of all zeros except value 1 at the $j$-th coordinate. We have

$$
\begin{aligned}
F(\bar{x}^{k+1}) - F(\bar{x}^k) &= F(x^{k+1}, \xi(x^{k+1})) - F(x^k, \xi^k) \\
&\leq F(x^{k+1}, \xi^k) - F(x^k, \xi^k) \\
&= \min_{\delta_j} \ h_j(x_j^k + \delta_j) + \nabla_j F(x^k)\delta_j + \frac{\eta_t\|\bar{a}_j\|^2}{2}\delta_j^2 \\
&\leq \min_{\delta_j} \ h_j(x_j^k + \delta_j) + \nabla_j F(x^k)\delta_j + \frac{M\eta_t}{2}\delta_j^2.
\end{aligned}
$$

Taking expectation of LHS and RHS w.r.t. $j$ yields the result. $\qquad\square$

Finally, notice that the function $g(z) = \frac{\eta_t}{2}\|z - b + y^t/\eta_t\|^2$ is locally Lipschitz-continuous with constant $L_g = \eta_t R_z$ for $z$ satisfying $\|z - b + y^t/\eta_t\| \leq R_z$, that is,

$$|g(z_1) - g(z_2)| \leq L_g\|z_1 - z_2\| \qquad (43)$$

for $\forall z_1, z_2$ with $\|z_1 - b + y^t/\eta_t\| \leq R_z$, $\|z_2 - b + y^t/\eta_t\| \leq R_z$, where $L_g$ is an upper bound on the magnitude of dual iterates $\|y^{t+1}\| = \|\eta_t(\bar{A}\bar{x}^k - b) + y^t\|$.

From simplicity of analysis, in the following, we slightly loosen upper bounds by setting constants $L_g \leftarrow \max(L_g, 1)$, $M \leftarrow \max(M, 1)$, $\theta \leftarrow \max(\theta, 1)$, such that $L_g, M, \theta \geq 1$. Then we are ready to prove the main theorem of this section.

**Theorem 5** (Linear Convergence). *The iterates $\{\bar{x}^k\}_{k=0}^\infty$ of RCD Algorithm satisfy*

$$\mathbb{E}[F(\bar{x}^{k+1})] - F^* \leq \left(1 - \frac{1}{n\gamma}\right)\left(\mathbb{E}[F(\bar{x}^k)] - F^*\right)$$

*where $F^*$ is the optimum of* (6) *and*

$$\gamma = \max\left\{16\eta_t M\theta(F^0 - F^*) \ , \ 2M\theta(1 + 4L_g^2) \ , \ 6\right\}.$$

*Proof.* Let $\bar{x}^* = \Pi_\mathcal{S}(\bar{x}^k)$ be the projection of $\bar{x}^k$ to the set of optimal solutions. From Lemma 4, we have

$$
\begin{aligned}
\mathbb{E}[F(\bar{x}^{k+1})] - F(\bar{x}^k) &\leq \frac{1}{n}\left(\min_\delta \ h(\bar{x}^k + \delta) + \langle\nabla F(\bar{x}^k), \delta\rangle + \frac{M\eta_t}{2}\|\delta\|^2\right) \\
&\leq \frac{1}{n}\left(\min_\delta \ h(\bar{x}^k + \delta) + F(\bar{x}^k + \delta) - F(\bar{x}^k) + \frac{M\eta_t}{2}\|\delta\|^2\right) \\
&\leq \frac{1}{n}\left(\min_{\alpha\in[0,1]} \ F(\bar{x}^k + \alpha(\bar{x}^* - \bar{x}^k)) - F(\bar{x}^k) + \frac{M\eta_t\alpha^2}{2}\|\bar{x}^* - \bar{x}^k\|^2\right) \\
&\leq \frac{1}{n}\left(\min_{\alpha\in[0,1]} \ -\alpha(F(\bar{x}^k) - F(\bar{x}^*)) + \frac{M\eta_t\alpha^2}{2}\|\bar{x}^* - \bar{x}^k\|^2\right),
\end{aligned}
$$
$$(44)$$

where the second and fourth inequality follow from the convexity of $F(\bar{x})$, and the third inequality follows from the fact that both $\bar{x}^*$ and $\bar{x}^k$ are feasible ($h(\bar{x}^*) = h(\bar{x}^k) = 0$). Now based on the error bound inequality (40), we discuss two cases.

**Case 1:** $4L_g^2\|\bar{A}\bar{x} - t^*\|^2 < (\bar{c}^T\bar{x} - s^*)^2$.

In this case, we have

$$\|\bar{x}^k - \bar{x}^*\|^2 \leq \theta \left(\|\bar{A}\bar{x}^k - t^*\|^2 + \|\bar{c}^T\bar{x}^k - s^*\|^2\right)$$
$$\leq \theta \left(\frac{1}{4L_g^2} + 1\right)(\bar{c}^T\bar{x}^k - s^*)^2 \leq 2\theta(\bar{c}^T\bar{x}^k - s^*)^2 \tag{45}$$

and

$$|\bar{c}^T\bar{x}^k - s^*| \geq 2L_g\|\bar{A}\bar{x}^k - t^*\| \geq 2|g(\bar{A}\bar{x}^k) - g(t^*)|.$$

Note in this case, $\bar{c}^T\bar{x}^k - s^*$ must be non-negative. Otherwise,

$$F(\bar{x}^k) - F^* = g(\bar{A}\bar{x}^k) - g(t^*) + (\bar{c}^T\bar{x}^k - s^*)$$
$$\leq |g(\bar{A}\bar{x}^k) - g(t^*)| - |\bar{c}^T\bar{x}^k - s^*|$$
$$\leq -\frac{1}{2}|\bar{c}^T\bar{x}^k - s^*| < 0,$$

leads to contradiction (since $\bar{x}^k$ is feasible, $F(\bar{x}^k)$ cannot be smaller than $F^*$). Therefore, we have

$$F(\bar{x}^k) - F^* = g(\bar{A}\bar{x}^k) - g(t^*) + \bar{c}^T\bar{x}^k - s^*$$
$$\geq -|g(\bar{A}\bar{x}^k) - g(t^*)| + \bar{c}^T\bar{x}^k - s^*$$
$$\geq \frac{1}{2}(\bar{c}^T\bar{x}^k - s^*). \tag{46}$$

Combining (44), (45), and (46), we have

$$\mathbb{E}[F(\bar{x}^{k+1})] - F(\bar{x}^k) \leq \frac{1}{n}\min_{\alpha\in[0,1]} -\frac{\alpha}{2}(\bar{c}^T\bar{x}^k - s^*) + \frac{2\eta_t M\theta\alpha^2}{2}(\bar{c}^T\bar{x}^k - s^*)^2$$
$$= \begin{cases} -1/(16\eta_t M\theta n) & , \ 1/(4\eta_t M\theta(\bar{c}^T\bar{x}^k - s^*)) \leq 1 \\ -\frac{1}{4n}(\bar{c}^T\bar{x} - s^*) & , \ o.w. \end{cases}$$

Furthermore, we have

$$-\frac{1}{16\eta_t M\theta n} \leq -\frac{1}{16\eta_t M\theta n(F^0 - F^*)}(F(\bar{x}^*) - F^*)$$

where $F^0 = F(\bar{x}^0)$, and

$$-\frac{1}{4n}(\bar{c}^T\bar{x} - s^*) \leq -\frac{1}{6n}(F(\bar{x}^k) - F^*)$$

since $F(\bar{x}^k) - F^* \leq |g(\bar{A}\bar{x}^k) - g(t^*)| + \bar{c}^T\bar{x}^k - s^* \leq \frac{3}{2}(\bar{c}^T\bar{x}^k - s^*)$. In summary, for Case 1 we obtain

$$\mathbb{E}[F(\bar{x}^{k+1})] - F^* \leq \left(1 - \frac{1}{n\gamma_1}\right)\left(\mathbb{E}[F(\bar{x}^k)] - F^*\right) \tag{47}$$

where

$$\gamma_1 = \max\left\{16\eta_t M\theta(F^0 - F^*), \ 6\right\}. \tag{48}$$

**Case 2:** $4L_g^2\|\bar{A}\bar{x}^k - t^*\|^2 \geq (\bar{c}^T\bar{x}^k - s^*)^2$.

In this case, we have
$$\|\bar{x}^k - \bar{x}^*\|^2 \leq \theta\left(1 + 4L_g^2\right)\|\bar{A}\bar{x}^k - t^*\|^2, \tag{49}$$

and by strong convexity of $g(z)$,

$$F(\bar{x}^k) - F^* \geq \bar{c}^T(\bar{x}^k - \bar{x}^*) + \nabla g(t^*)^T\bar{A}(\bar{x}^k - \bar{x}^*) + \frac{\eta_t}{2}\|\bar{A}\bar{x}^k - t^*\|^2.$$

Adding inequality $0 = h(\bar{x}^k) - h(\bar{x}^*) \geq \langle \rho^*, \bar{x}^k - \bar{x}^*\rangle$ for some $\rho^* \in \partial h(\bar{x}^*)$ to the above gives

$$F(\bar{x}^k) - F^* \geq \frac{\eta_t}{2}\|\bar{A}\bar{x}^k - t^*\|^2 \tag{50}$$

since $\rho^* + \bar{c} + \nabla g(t^*)^T \bar{A} = \rho^* + \nabla F(\bar{x}^*) = 0$. Combining (44), (49), and (50), we obtain

$$\mathbb{E}[F(\bar{x}^{k+1})] - F(\bar{x}^k) \leq \frac{1}{n} \min_{\alpha \in [0,1]} -\alpha(F(\bar{x}^k) - F^*) + \frac{M\theta(1 + 4L_g^2)\alpha^2}{2}\left(F(\bar{x}^k) - F^*\right)$$

$$= -\frac{1}{2M\theta(1 + 4L_g^2)n}\left(F(\bar{x}^k) - F^*\right) \tag{51}$$

Combining results of Case 1 (47) and Case 2 (51), and taking expectation on both sides w.r.t. the history leads to the result (19). □

We then bounds the number of iterations required to achieve $\epsilon$ sub-optimality with high probability $1 - p$ by the following corollary.

**Corollary 2** (Inner Iteration Complexity). *To guarantee*

$$F(\bar{x}^k) - F^* \leq \epsilon \tag{52}$$

*with probability $1 - p$, it suffices running RCD Algorithm 2 for*

$$k \geq \gamma n \log\left(\frac{F(\bar{x}^0) - F^*}{\epsilon p}\right)$$

*iterations, where $\gamma$ is constant defined in Theorem 5.*

*Proof.* We use the Theorem 1 of [26] to transfer the linear convergence in expectation (19) into iteration complexity. To do this, we express (19) in the form

$$\mathbb{E}[F(\bar{x}^{k+1})] - F^* \leq \left(1 - \frac{1}{c}\right)\left(\mathbb{E}[F(\bar{x}^k)] - F^*\right),$$

with $c = \gamma n$, and then apply the theorem to show that $c\log(\frac{1}{\epsilon p})$ updates suffice to guarantee $F(\bar{x}^k) - F^* \leq \epsilon$ with probability $1 - p$. □

To relate the solution quality of sub-problem (6) to the outer proximal iterations (5), we need to bound not only the function difference in primal but also the distance to the exact solution $y^{t+1} = \mathbf{prox}_{\eta_t g}(y^t)$ to the proximal update (5). To achieve this, we transfer the bound on $F(\bar{x}^k) - F^*$ to that on $\|y(\bar{x}^k) - y^{t+1}\|$.

**Corollary 3.** *To guarantee*

$$\|y(\bar{x}^k) - y^{t+1}\| \leq \epsilon_0 \tag{53}$$

*with probability $1 - p$, it suffices running RCD for*

$$k \geq 2\gamma n \log\left(\sqrt{\frac{2\eta_t(F(\bar{x}^0) - F^*)}{p}}\frac{1}{\epsilon_0}\right)$$

*iterations.*

*Proof.* Given the primal iterate $\bar{x}^k$, the corresponding dual iterate $y(\bar{x}^k)$ is maintained through (7), written as

$$y(\bar{x}^k) = \eta_t(\bar{A}\bar{x}^k - b) + y^t.$$

Therefore,

$$\|y(\bar{x}^k) - y^{t+1}\| = \|\bar{A}(\bar{x}^k - \bar{x}_{\mathcal{S}}^k)\|. \tag{54}$$

To bound (54) by the function value difference, note that

$$F(\bar{x}^k) - F(\bar{x}_{\mathcal{S}}^k) = \langle \nabla F(\bar{x}_{\mathcal{S}}^k), \bar{x}^k - \bar{x}_{\mathcal{S}}^k \rangle + \frac{1}{2}(\bar{x}^k - \bar{x}_{\mathcal{S}}^k)^T \nabla^2 F(\bar{x}_{\mathcal{S}}^k)(\bar{x}^k - \bar{x}_{\mathcal{S}}^k)$$

and since

$$0 = h(\bar{x}^k) - h(\bar{x}_{\mathcal{S}}^k) \geq \langle \rho^*, \bar{x}^k - \bar{x}_{\mathcal{S}}^k \rangle$$

$(\rho^* \in \partial h(\bar{x}_S^k)$ is the unique subgradient at optimal defined in (35)), together we get

$$F(\bar{x}^k) - F(\bar{x}_S^k) \geq \frac{1}{2}(\bar{x}^k - \bar{x}_S^k)^T \nabla^2 F(\bar{x}_S^k)(\bar{x}^k - \bar{x}_S^k) = \frac{\eta_t}{2}\|\bar{A}(\bar{x}^k - \bar{x}_S^k)\|^2,$$

which, combined with (54), leads to the bound

$$\|y(\bar{x}^t) - y^{t+1}\| \leq \sqrt{2\eta_t\left(F(\bar{x}^k) - F(\bar{x}_S^k)\right)}.$$

Therefore, to guarantee $\|y(\bar{x}^k) - y^{t+1}\| \leq \epsilon_0$, it suffices to have $F(\bar{x}^k) - F(\bar{x}_S^k) \leq \frac{\epsilon_0^2}{2\eta_t}$, which can be achieved with high probability $1 - p$ by running RCD Algorithm 2 for

$$k \geq \gamma n \log\left(\frac{2\eta_t(F(\bar{x}^0) - F^*)}{\epsilon_0^2 p}\right) = 2\gamma n \log\left(\sqrt{\frac{2\eta_t(F(\bar{x}^0) - F^*)}{p}}\frac{1}{\epsilon_0}\right) \tag{55}$$

according to Corollary 2. $\qquad\square$

### A.3 Overall Iteration Complexity of AL-CD

This section combines the linear convergence of Augmented Lagrangian (AL) and Coordinate Descent (CD) to give an overall iteration complexity that bounds the number of RCD updates required for AL-CD to find an LP solution of $\epsilon$ precision.

The first key lemma bounds the approximation error incurred in the outer iterates when solving inner sub-problems in an inexact fashion.

**Lemma 5** (Inexact Proximal Map). *Suppose, for a given dual iterate $y^t$, each sub-problem (6) is solved inexactly s.t.*

$$\|\hat{y}^{t+1} - \mathbf{prox}_{\eta_t g}(y^t)\| \leq \epsilon_0. \tag{56}$$

*Then let $\{\hat{y}^t\}_{t=1}^{\infty}$ be the sequence of iterates produced by inexact proximal updates and $\{y^t\}_{t=1}^{\infty}$ as that generated by exact updates. After $t$ iterations, we have*

$$\|\hat{y}^t - y^t\| \leq t\epsilon_0. \tag{57}$$

*Proof.* By the non-expansiveness of proximal operation,

$$\begin{aligned}
\|\hat{y}^{t+1} - y^{t+1}\| &\leq \|\hat{y}^{t+1} - \mathbf{prox}_{\eta_t g}(\hat{y}^t)\| + \|\mathbf{prox}_{\eta_t g}(\hat{y}^t) - y^{t+1}\| \\
&\leq \epsilon_0 + \|\mathbf{prox}_{\eta_t g}(\hat{y}^t) - \mathbf{prox}_{\eta_t g}(y^t)\| \\
&\leq \epsilon_0 + \|\hat{y}^t - y^t\|.
\end{aligned}$$

Recursively applying the above inequality leads to the conclusion (57). $\qquad\square$

Note the above implies that, if an exact AL method performs $t$ outer iterations to achieve $\epsilon$-precise solution, then solving each subproblem with precision $\epsilon_0 = \epsilon/t$ makes only an additional $\epsilon$ approximation error in the overall result. This insight turns out to give the following main theorem.

**Theorem 6** (Iteration Complexity). *Denote $\{\hat{y}^t\}_{t=1}^{\infty}$ as the sequence of iterates obtained from inexact dual proximal updates and $\{y^t\}_{t=1}^{\infty}$ as that generated by exact updates. To guarantee $\|\hat{y}^t - \hat{y}_{S_*}^t\| \leq 2\epsilon$ with probability $1 - p$, it suffices to run Algorithm 1 for*

$$T = (1 + \frac{1}{\alpha})\log\left(\frac{LR}{\epsilon}\right) \tag{58}$$

*outer iterations with $\eta_t = (1 + \alpha)L$, and solve each sub-problem (6) by running Algorithm 2 for*

$$k \geq 2\gamma n \left(\log\left(\frac{\omega}{\epsilon}\right) + \frac{3}{2}\log\left((1 + \frac{1}{\alpha})\log\frac{LR}{\epsilon}\right)\right) \tag{59}$$

*inner iterations, where $\omega = \sqrt{\frac{2(1+\alpha)L(F^0 - F^*)}{p}}$, $R = \|\Delta^0\|$.*

*Proof.* Since
$$\|\hat{y}^t - \hat{y}^t_{S_*}\| \leq \|\hat{y}^t - y^t_{S_*}\| \leq \|y^t - y^t_{S_*}\| + \|\hat{y}^t - y^t\|,$$
to guarantee $\|\hat{y}^t - \hat{y}^t_{S_*}\| \leq 2\epsilon$, it suffices to let $\|y^t - y^t_{S_*}\| < \epsilon$ and $\|\hat{y}^t - y^t\| < \epsilon$, where the former can be guaranteed as long as the number of outer iterations
$$T = (1 + \frac{1}{\alpha}) \log \left( \frac{L\|\Delta^0\|}{\epsilon} \right)$$
by Corollary 1. To ensure $\|\hat{y}^t - y^t\| < \epsilon$, according to Lemma 5, it suffices to solve each proximal subproblem to precision $\epsilon_0 = \epsilon/T$. To guarantee that the $T$ subproblems are all solved to precision $\epsilon_0 = \epsilon/T$ with probability $1 - p$, we require each of them to hold with probability $1 - p/T$ independently, which can be guaranteed by running RCD on each subproblem for
$$k \geq 2\gamma n \log \left( \sqrt{\frac{2(1+\alpha)L(F_t(\bar{x}^0) - F_t^*)}{p}} \frac{T^{3/2}}{\epsilon} \right)$$
inner iterations (Corollary 3), where $F_t(\bar{x})$ denotes the objective of $t$-th subproblem. To remove the dependency of $k$ on $t$, we bound the term $F_t(\bar{x}^0) - F_t^*$ by $F^0 - F^*$, where $F^* \leq F_t^*$ is a lower bound on the optimal function value of subproblem, which exists as long as the original LP is bounded below, and $F^0 \geq F_t(\bar{x}^0)$ is a bound on the initial function value of each sub-problem, which exists as long as each subproblem is initialized by the solution of previous subproblem, and each subproblem is solved with precision $\epsilon_0 = \epsilon/T$. Then to guarantee the above inequality, it suffices to have
$$k \geq 2\gamma n \left( \log \left( \frac{\omega}{\epsilon} \right) + \frac{3}{2} \log \left( (1 + \frac{1}{\alpha}) \log \frac{LR}{\epsilon} \right) \right),$$
where $\omega = \sqrt{\frac{2(1+\alpha)L(F^0 - F^*)}{p}}$, $R = \|\Delta^0\|$. $\qquad\square$

# B  Appendix-B. Data Statistics

All data sets for experiments of L1-regularized SVM can be found in the LIBSVM dataset repository, where the data set *cod-rna.rf* uses $D = 5000$ Fourier Random Features [32, 33] to approximate the effect of Gaussian RBF kernel. We choose $\lambda = 1$ for all L1-regularized SVM problems except for *cod-rna.rf* we use $\lambda = 10$ to increase the primal sparsity. The data sets *textmine*, *E2006* for Sparse Inverse Covariance Estimation are also obtained from LIBSVM dataset repository, while the *micromass*, *dorothea* are taken from UCI Machine Learning repository. For Sparse Inverse Covariance Estimation, we excluded features of frequency less than 10. The *ocr* data set is taken from `http://ai.stanford.edu/~btaskar/ocr/`. For Non-negative Matrix Factorization, we set the matrix approximation tolerance to be 0.01 times number of samples.

Table 4: Data Statistics for L1-SVM

| Data set | #Samples | #Features | NNZ | #class | $n_b$ | $m_I$ |
|---|---|---|---|---|---|---|
| rcv | 15564 | 47236 | 1028284 | 53 | 4833738 | 778200 |
| news | 15935 | 62061 | 1272569 | 20 | 2498415 | 302765 |
| sector | 6412 | 55197 | 1045412 | 105 | 11597992 | 666848 |
| mnist | 60000 | 780 | 8994156 | 10 | 75620 | 540000 |
| cod-rna.rf | 59535 | 5000 | 297675000 | 2 | 69537 | 59535 |
| viecle | 78823 | 100 | 7882300 | 3 | 79429 | 157646 |
| real-sim | 72309 | 20958 | 3709083 | 2 | 114227 | 72309 |

Table 5: Data Statistics for Sparse Inverse Covariance Selection

| Data set | #Samples | #Features | NNZ | $n_b$ | $m_I$ | $m_E$ | $n_f$ |
|---|---|---|---|---|---|---|---|
| textmine | 21519 | 30438 | 2283179 | 60876 | 60876 | 43038 | 43038 |
| E2006 | 16087 | 27917 | 19640157 | 55834 | 55834 | 32174 | 32174 |
| dorothea | 800 | 23616 | 463088 | 47232 | 47232 | 1600 | 1600 |

Table 6: Data Statistics for Convex Nonnegative Matrix Factorization

| Data set | #Samples | #Features | NNZ | $n_b$ | $m_I$ |
|---|---|---|---|---|---|
| micromass | 931 | 1,299 | 106,292 | 2,896,770 | 4,107,438 |
| ocr | 52,152 | 127 | 1,466,486 | 6,639,433 | 13,262,864 |