[Reviews · NeurIPS 2015]

Submitted by Assigned_Reviewer_1

In this paper, an approach that combines Augmented Lagrangian (AL) and Coordinate Descent (CD) methods are proposed to solve large scale sparse linear programming. The overall iteration complexity and the bounds of total number of CD are discussed for finding an eps-accurate solution. Moreover, an active set strategy is introduced for primal and dual AL-CD to speed up the convergence. The AL-CD method provides an alternative to interior point and simplex methods when it is infeasible to a linear system exactly. The proposed method is very interesting and useful in practice. The paper is organized well and clearly presented all the main results. Two questions:

(1) Page 4, line 182. g(y) is nonsmooth. It might be better to clearly explain why. g(y) in (2) is linear, which is continuous. It is nonsmooth, is it because the added condition, if y is infeasible, and take g(y)=\infty? Also, line 183, smoothed dual of (5)? What does this mean? The objective function of (6) is smooth? It might be better to clearly explain them in the paper.

(2) Page 6, line 285-286, (6) is strongly convex when restricted to a constant subspace? What is the constant subspace?

Other problems:

- Typo, page 2, line 083, O(nnz(A), missing the right bracket.

- Typo, line 207. the part of variables in \xi can be minimized in a closed-form as...

- Page 6, line 277, missing the period . at the end of the sentence.

- Page 6, line 282, typo, showon-> shown.

- Page 6, line 322, what is the operator "prox"?

- Page 5, Algorithm 3, until, d^*_{A^{t,k}}, A should be \mathcal{A}.

- Page 8, line 386, U,V is sparse, "is" should be "are".

- Section 5, it might be better to add the primal-dual gap of AL-CD method, which is a very important criteria to check the convergence quality.

- Page 6, the convergence analysis discussion. The proof of Theorem 1 is in Appendix A Theorem 5, so it might be better to list in the paper which one is the proof of Theorem 1. Similarly for Theorem 2 and so on.
Summary: In this paper, an approach that combines Augmented Lagrangian (AL) and Coordinate Descent (CD) methods are proposed to solve large scale sparse linear programming.

The proposed method is very interesting and useful in practice. The paper is organized well and clearly presented all the main results. It only has a few questions.

Submitted by Assigned_Reviewer_2

*The problem is definitely relevant and the overall treatment rigorous and thorough. I liked reading the paper, it is a valuable contribution to the literature.

*The experimental sections are a bit underwhelming. While CPLEX/Gurobi are good baselines for the most general LP instantiations, for specific instances considered here, is CPLEX really the best one can do? Also, why not throw in a SGD with projection? What seemed confusing is the omission of more standard LPs in the head-to-head comparisons, SVMs etc are fine to include but should be secondary to an experimental design that supports the message that the algorithms presented here are powerful options for *sparse LPs*, not merely for models where several geometric/specialized methods are available.

*For practical settings (which seems to be the motivation here), it will be useful to have expanded versions of Tables 1--3 with additional baselines demonstrating quality of solutions not in the asymptotic sense, but for a fixed budget. To me, Section 3.4 is fairly mechanical and can be easily transferred to the supplement to make space for expanding Section 4.

*The supplement is not easy to read standalone unless one is intricately familiar with the papers referred there. I had to spend significant time digging up the references to follow the reasoning in the proofs. If accepted, it will help much if the authors spend a little more effort in making the appendix self contained.

Summary: This paper presents methods to solve large scale LPs in ML and other applications approximately when the constraint matrix is sparse. The solution has runtime complexity which depends on the

number of non-zeros in the constraint matrix (and also level of precision desired). The scope of the paper is the ground between interior point methods and cheaper methods; the authors seek to derive a scheme which obtains better quality solutions (than SGD) while offering efficiency benefits over interior point/simplex. A randomized coordinate descent method is described and fits nicely relative to recent work in ML (and a renewed interest in optimization in CD methods). Convergence properties are analyzed and a small set of experiments are presented.

Submitted by Assigned_Reviewer_3

It will be nice to make this solver be public available. Besides, I don't see the KKT conditions used in the article. The invalidation of KKT conditions should suggest the active set selection.
Summary: This is nice to revisit the linear program and adding on the "sparsity" constraint. The proposed algorithm tries to combine the modern optimization techniques to solve the model. It can be applied to many machine learning problems.

Submitted by Assigned_Reviewer_4

This paper considers the use of augmented Lagrangian or the method of multipliers to solve the linear program, attempting to address the scalability issue of classical linear program methods like interior point methods and simplex methods for large scale high dimensional machine learning problems. The resulting augmented Lagrangian method basically reduces to two steps which update primal and dual variables alternatingly. The primal variable update is to solve a bound-constrained quadratic problem which involves high computational effort.

To solve the primal subproblem, the authors proposed to use another optimization algorithm like randomized block coordinate descent to solve it. Along with the outerloop, the proposed algorithm is a double-loop algorithm.

The

authors also consider several improvements, e.g., active-set. Simulation results show that the proposed method is faster than other LP solvers.

Overall, it is a good attempt to solve large scale LP using a recent method. However, in addition to the use in LP, the algorithm has already been studied in the literature. Moreover,

1. The proposed algorithm is not novel and has been used in other problems.

Separable approximations and decomposition methods for the augmented lagrangian. R. Tappenden, P. Richtarik, and B. Buke, 2013.

2. Instead of solving the subproblem (6) using other optimization methods, an more efficient algorithm can randomly sweep some coordinates at each iteration, which was proposed in couple of papers:

Parallel direction method of multipliers, Huahua Wang and Arindam Banerjee, NIPS, 2014.

Stochastic Primal-Dual Coordinate Method for Regularized Empirical Risk Minimization, ArXiv

These algorithms should lead to more efficient algorithm to solve LP, although the convergence analysis on LP is missing.

---------------------------

The author's response addressed some questions I had. As a whole, the paper is well written and made a contribution in the use of state-of-the-art method to solve LP, but it is not novel enough.

Summary: Using augmented Lagrangian along with randomized coordinate descent to solve LP is a good attempt.

The idea and the algorithm are not novel enough.

Submitted by Assigned_Reviewer_5

In this paper, the authors propose a method which combines augmented Lagrangian and coordinate descent techniques for solving large scale linear programs with a sparse constraint matrix.

Pros and cons --------------

+ Overall, this is a strong contribution. In the introduction, the authors did a good job motivating their paper and covering the related work. Coordinate descent is a good fit for solving the inner problem.

+ The authors give iteration complexity guarantees for the entire algorithm, even if the inner problem is solved inexactly.

+ I particularly appreciated that the experiments cover three representative linear programs in machine learning: l1-regularized multiclass SVM, NMF and sparse covariance matrix estimation.

- The projected Newton-CG (PNCG) method is not covered in the experiments.

- The experiments are rather short and only indicate timing results. I would

have preferred to see more experiments and move the PNCG part to the supplementary material.

- There is no discussion whatsoever on how the parameter \eta_t was chosen. If the paper is accepted, it is very important to address this issue in the camera-ready version.

- Same problem with regularization parameters in the experiments.

- While much better than interior point and simplex methods, the timing results

remain too large for practical use. In practice, one can add a little bit of l2 regularization to the problem (i.e., replace l1 regularization by elastic-net).

Then the problem can be solved by modern solvers such as SDCA, which should be orders of magnitude faster than results shown in this paper.

Detailed comments ------------------

* The authors claim in several places in the paper that the conditioning of the problem doesn't worsen with the number of iterations. It would be interesting to briefly explain why.

* line 101: non-negative constraint -> non-negativity constraint

* line 176: g(y) is already defined in (2). Here you are overloading the definition so that g(y) becomes infinity if y is infeasible. I hadn't realized this at first glance and so (5) made no sense. Instead of overloading g(y), you could define a new function \tilde{g}(y) to make it more explicit.

* problem (5): you didn't define \eta_t

* line 230: you already defined \hat{F}(x) in (9)

* line 236: add citation for generalized Hessian

* line 259: clever -> efficient

* Table 1-3: Barrier = IPM?
Summary: A pretty strong contribution but experiments could have been better.

Author Feedback
Author rebuttal: We are thankful to all reviewers for their careful and constructive comments.

To Reviewer_5:

1. Comparison to [2].

The PDMM [2] (or other variants of multi-block ADMM) introduces cheaper procedure in the update of primal variables to avoid a double-loop algorithm. However, it has very conservative updates on the dual variables and requires far more iterations for convergence. In particular, the step size of dual update in [2] is inversely proportional to #coordinates, which can be up to 10^7 in our LP instance, leading to a step size of 10^{-7}.

In comparison, ALM reduces number of outer iterations by putting more load on solving the bound-constrained quadratic sub-problem, where, however, a relatively stable active set of variables and constraints can be identified to reduce problem size significantly. In our experiment the ALM typically converges within 100 iterations.

2. Comparison to [1].

Although "augmented lagrangian" is in the title of [1], the paper does not analyze ALM itself. Instead it focuses on the comparison of DQAM and PCDM in solving the primal sub-problem of ALM. Furthermore, the linear convergence result in [1] requires strong convexity of the augmented lagrangian (AL) function, which does not hold in the LP case.

On the other hand, the analysis in our paper treated AL-CD as a whole and provide linear-type of convergence without requiring strong convexity of the AL function.

[1] Separable approximations and decomposition methods for the augmented lagrangian. R. Tappenden, P. Richtarik, and B. Buke, 2013.
[2] Parallel direction method of multipliers, H. Wang and A. Banerjee, NIPS 2014.

To Reviewer_2:

1. Choice of \eta_t.

As stated in Sec. 4.1, we employ a two-phase strategy where in the first phase \eta_t is fixed, while in the second phase, we double \eta_t whenever the primal-infeasibility does not decrase sufficiently.

For all experiments, we set the initial \eta_t=1. We will state this more clearly in the experimental section.

2. Choice of regularization parameter.

In all experiments of L1-regularized multiclass SVM we set \lambda=1. We will add more details on how the LP is formed in the appendix.

3. Elastic-net regularized SVM.

We agree that adding an additional L2-regularizer to the L1-SVM problem will make the problem easier. Actually, this is equivalent to performing only "one" outer iteration of the proximal update (5) in the primal with initialization at 0, which without doubt will be an order of magnitude faster.

4. Why does conditioning of subproblem in ALM not worsen with the number of iterations?

The main factor determining iteration complexity for solving each sub-problem is the largest/smallest eigenvalues of Hessian matrix (of function (31) in the appendix), which does not change over iterations. This is in contrast to the matrix of linear system solved in IPM, which change dramatically when approaching the boundary.

(We appreciate other detailed comments and will improve write-up based on those.)

To Reviewer_4:

1. Comparison with SGD.

We agree there should be a comparison of Al-CD and SGD with fixed budget, but this can be only done under context of specific application, since when the problem is solved loosely, the solution is usually not feasible and comparing objective functions of two infeasible solutions does not make sense.

2. Experiments on other sparse LPs.

We include LP instances in Machine Learning to fit better to the audience of the conference, but there are indeed many standard sparse Linear Programs outside Machine Learning. If time permits, we will add experiments on more sparse LPs into appendix.

To Reviewer_6:

1. Clarification about eq (5), (6).

As pointed out by Reviewr_2, g(y) should be \tilde{g}(y) and, yes, it is non-smooth because it is \infty when y is infeasible.

We call (6) "smoothed dual" because the penalty incurs for infeasibility becomes quadratic instead of \infty as in (1). We will use better terminology to avoid confusion.

2. What is the "constant subspace" in line 285, 286 ?

It is the orthogonal space of Nullspace of \bar{A}, as defined in Lemma 1 in the appendix.

3. We appreciate other detailed comments and will be improving based on these.

To Reviewer_1:

1. The use of KKT conditions in finding active set.

We use the magnitude of projected gradient to select active set, which is equivalent to using KKT condition of the bound-constrained problem (9).

To Reviewer_3:

1. The lack of duality gap for checking optimality.

We agree duality gap is a good measure for optimality. However, primal, dual feasibility also implies optimality in our case. A dual feasible y^{t+1}(x^{t+1}) means the projected gradient of (9) is zero at x^{t+1}, while a primal feasibile x^{t+1} means y^{t+1}-y^{t}=0, which implies g(y^{t+1}) is optimal.